# Does Time to Adjuvant Chemotherapy after Radical Cystectomy Affect Survival in Muscle Invasive Bladder Cancer? A Systematic Review

**DOI:** 10.3390/cancers14225644

**Published:** 2022-11-17

**Authors:** Shane Kronstedt, Sai Krishnaraya Doppalapudi, Joseph Boyle, Kevin Chua, Thomas L. Jang, Giovanni E. Cacciamani, Saum Ghodoussipour

**Affiliations:** 1Division of Urology, Rutgers Robert Wood Johnson Medical School, New Brunswick, NJ 08901, USA; 2Cancer Institute of New Jersey, New Brunswick, NJ 08903, USA; 3Catherine and Joseph Aresty Department of Urology, USC Institute of Urology, Keck School of Medicine, University of Southern California, Los Angeles, CA 90033, USA

**Keywords:** bladder cancer, cystectomy, adjuvant chemotherapy

## Abstract

**Simple Summary:**

With enhanced recovery after surgery protocols, it is now possible to decrease a patient’s length of stay in the hospital and administer adjuvant chemotherapy earlier than before. Our study analyzed 7056 patients who underwent a radical cystectomy and subsequent adjuvant chemotherapy to determine if an earlier timing of adjuvant chemotherapy would affect survival outcomes. We found that earlier time to adjuvant therapy showed a survival benefit, and a benefit persisted regardless of timing when compared to controls.

**Abstract:**

(1) Purpose: To assess the survival benefit for different times to adjuvant chemotherapy after a radical cystectomy. (2) Materials and Methods: We systematically searched PubMed^®^, Cochrane Central^®^, Scopus^®^, and Web of Science^®^ library databases for original articles that looked at timing to adjuvant chemotherapy after radical cystectomy. Primary endpoints were five-year survival, progression free survival, and overall survival. Available multivariable hazard ratios and corresponding 95% CIs were included in the qualitative analysis. The risk of bias was completed for nonrandomized studies. (3) Results: Using PRISMA guidelines, our electronic search resulted in a total of 1862 records. After a detailed review, we selected four studies that addressed the impact of the timing of adjuvant chemotherapy for patients who underwent radical cystectomy. (4) Conclusion: A survival benefit was seen with an earlier administration of adjuvant chemotherapy, albeit a benefit persists for delayed chemotherapy post-radical cystectomy. A safe and ethical approach at this time would be to administer adjuvant chemotherapy as early in the postoperative period as possible, given the known survival benefit of such therapy (9–11% absolute survival benefit at five years).

## 1. Introduction

The standard of care management of muscle-invasive bladder cancer (MIBC) is neoadjuvant cisplatin-based chemotherapy followed by radical cystectomy and pelvic lymph node dissection. The utilization of neoadjuvant chemotherapy (NAC) in this patient population was justified in a randomized controlled trial (RCT), which showed improved median overall survival (OS) (77 months vs. 46 months, *p* = 0.06) and pathological downstaging (pT0 in 38% vs. 15%) in those receiving cisplatin-based chemotherapy followed by surgery vs. surgery alone [1]. These findings were further substantiated by a meta-analysis of 11 trials, which showed a 5% and 9% absolute improvement in 5-year OS and disease-free survival (DFS), respectively [2]. Despite this evidence, a small proportion of eligible patients receive NAC, with contemporary estimates being approximately 27% [3]. Providers may opt for upfront cystectomy and adjuvant chemotherapy (AC) if needed, as delays to surgical consolidation lead to higher overall mortality (HR without NAC 1.34 [1.03–1.76]; HR with NAC 1.63 [1.06–2.52]) [4]. Recent data also support the use of immunotherapy after RC in high-risk patients regardless of preoperative chemotherapy use [5].

In a meta-analysis of 9 RCTs, including a total of 945 patients who had surgery without NAC, receipt of AC improved OS (HR 0.77, *p* = 0.049) and DFS (HR 0.66, *p* = 0.014). Improvements in DFS were most apparent in patients with node-positive disease [6]. A more recent meta-analysis of 10 RCTs published in 2021 showed an absolute survival benefit of 6% at five years in patients receiving AC (9% when adjusting for confounding variables) [7]. Unfortunately, due to the significant morbidity of radical cystectomy, many patients will experience considerable delays in systemic treatment. Implementing enhanced recovery after surgery (ERAS) protocols have recently improved perioperative care after RC. ERAS protocols are multimodal pathways that optimize all elements of perioperative care. This includes preoperative, intraoperative, and postoperative modifications to enhance recovery and reduce stress following surgery. These protocols have reduced postoperative length of stay without change in complication or readmission rates, now making it possible to administer AC earlier than usual. The impact of ERAS on time to AC and survival is not yet clear [8,9]. This systematic review aims to determine if earlier administration of adjuvant chemotherapy can significantly improve survival trends in this patient population.

## 2. Materials and Methods

For this systematic review, we followed the PRISMA (Preferred Reporting Items for Systematic Review and Meta-Analyses) statement. PubMed^®^, Cochrane Central^®^, Scopus^®^, and Web of Science^®^ library databases were queried for full-text articles that addressed whether the timing of AC affected survival in patients with MIBC after a radical cystectomy and pelvic lymph node dissection. Search terms included “adjuvant therapy for muscle invasive bladder cancer” “adjuvant chemotherapy for muscle invasive carcinoma of the bladder” OR “adjuvant for muscle invasive urothelial carcinoma.” A secondary search was conducted to analyze mortality and timing of the adjuvant immunotherapy (IO) nivolumab, adding 397 records. Search terms included “adjuvant therapy for MIBC” OR “nivolumab adjuvant bladder” OR “nivolumab adjuvant cystectomy.” This study is registered with PROSPERO (https://www.crd.york.ac.uk/prospero/display_record.php?ID=CRD42022316360, accessed on 15 June 2022).

Two paired investigators (SK, JB) independently screened 1862 articles, focusing on studies reporting survival outcomes and their relation to timing to adjuvant therapy after radical cystectomy. Any disagreements about eligibility were resolved by discussion among the two investigators or finally resolved by the senior author (SG) until a consensus was reached. We included studies reporting the impact of time to adjuvant therapy on the outcomes of interest. A total of 1862 records were identified using the aforementioned search criteria. Inclusion criteria included adults 18 years and older diagnosed with MIBC (as diagnosed using recognized diagnostic criteria) who received a radical cystectomy and subsequent adjuvant therapy. Exclusion criteria included trials comparing NAC and AC alone, trials lacking comparison of timelines to adjuvant therapy (e.g., greater or less than 45 days postoperatively), trials comparing against relapse or salvage therapy alone, and non-English publications. No limits based on the year of publication, study design, or additional patient-specific demographics (other than those imposed by the studies themselves) were implemented in this systematic analysis. Four full-text articles were selected for this systematic review using PRISMA guidelines after implementing the exclusion and inclusion criteria (Figure 1).

When an institution published multiple series with entire overlapping surgical periods on the same topic, we considered the latest and largest published series. However, studies from the same institution with entire or partly overlapping surgical periods but evaluating different study populations were included in the analysis. To assess the overlapping studies’ bias, meta-regression models were performed. Studies analyzing the same national databases were excluded due to the high risk of overlapping data. The PICOTS format (Population, Intervention, Comparator, Outcome, Timing and Setting) fully summarizes our research and analysis strategy for evaluating the long-term oncologic and survival outcomes. Two paired investigators (SK, JB) independently weighed the risk of bias for all studies according to the Cochrane Handbook for Systematic Reviews of Interventions for including nonrandomized studies [10]. In consideration of the design of studies, we similarly examined the risk of preassigned confounders identified as the main prognostic factors at the time of RC. We then reviewed the articles that adjusted for the effects of age, gender, race, comorbidity, clinical or pathological stage, and positive surgical margins within their Cox proportional hazards regression models when available. All articles were categorized according to the Oxford Levels of Evidence Working Group 2011 levels of evidence for therapy studies according to the GRADE (Grading of Recommendations Assessment, Development and Evaluation) approach [11,12].

## 3. Results

Of the four articles selected, all four were retrospective studies (Table 1, Table 2 and Table 3). Of the four retrospective studies, two derived data from the National Cancer Database (NCDB), one from the Ontario Cancer Registry (OCR), and one from the Retrospective International Study of Invasive/Advanced Cancer of the Urothelium (RISC) database. There was a potential overlap of patient data in the two NCDB studies patients were analyzed from 2004 to 2014 in one study (*n* = 284) and 2006–2013 in the other (*n* = 3590); however, they measured different outcomes and selected patients differently.

In 2014, the Queen’s University Cancer Research Institute in Ontario, Canada, and the Princess Margaret Cancer Centre in Toronto, Canada, conducted a retrospective review using the Ontario Cancer Registry database to compare the timing of AC and its effect on survival. Cystectomy was performed in 2944 patients, and they were separated into NAC overall, AC overall, AC < 12 weeks, and AC > 12 weeks (23% [*n* = 120]). The majority of patients had pT3-pT4 disease (82%; *n* = 446), and N+ disease (67%; *n* = 363). The mean time to AC (TTAC) was 10 weeks; individuals in the > 12 weeks group had inferior OS (HR 1.28 [1.00–1.62]) and cancer-specific survival (HR 1.30 [1.00–1.69]) than those who had a shorter time to AC after surgery. Cisplatin-based regimens were most commonly used (82%), to include gemcitabine-cisplatin (54% [*n* = 166]), and methotrexate, vinblastine, doxorubicin, cisplatin (21% [*n* = 66]). Carboplatin-based therapy was used in 17% (*n* = 43). Information on the dose, number of cycles, and patient-level chart information in 54% of cases (those not treated at a comprehensive cancer center) were not provided [13].

A recent multi-institutional, international study was conducted by Grunewald et al., in 2020 using the RISC database to analyze the impact of the timing of AC on survival after RC [14]. Ninety days (approximately 12 weeks) was used as the cutoff for the surgery to AC interval. Of the 238 eligible patients, 76.1% (*n* = 181) had pT3-pT4 disease, and 66.4% (*n* = 158) had N+ disease. Median TTAC was 57 days (IQR 32.8); there was no association between TTAC and any patient or tumor-related factors. An analysis of OS showed a trend toward statistical significance with decreased TTAC (HR 1.004 [0.9997–1.0084]; *p* = 0.071). When a dichotomous analysis was performed with patients separated by TTAC AC < 90 days and > 90 days, trends toward better outcomes were seen, but differences in OS (*p* = 0.438) or progression-free survival (PFS; *p* = 0.056) were not noted to be statistically significant [14].

The Department of Urology and Hematology at the University of Pennsylvania conducted a retrospective review of 3590 patients (1009 who received multiagent AC) registered in the NCDB from 2006 to 2013. Patients were included if they had locally advanced pT3-T4 and/or pN+ MIBC. They were separated into four cohorts: no AC (*n* = 2581), AC 31–60 days (*n* = 538), 61–90 days (321), and AC 91–180 days (*n* = 150). The overall median survival of all AC-treated patients post-RC was 35.8 months compared to 26.9 in RC alone. All three treatment groups showed a statistically significant decrease in mortality (compared to RC patients who did not receive AC) in post-RC patients who received AC 31–60 days (HR 0.60 [0.52–0.69]; *p* < 0.001), AC 61–90 days (HR 0.62 [0.53–0.74]; *p* < 0.001), and AC 91–180 days (HR 0.69 [1.05–1.52]; *p* = 0.020). Patients with an increased likelihood to receive early AC (53% [*n* = 538] at 31–60 days and 47% [*n* = 471] at 61–180 days post-RC) had a median income >$62,999 (OR 1.83 [1.06–3.14]; *p* = 0.028) and pN1 disease (OR 1.58 [1.11–2.25]; *p* = 0.011) or pN2+ disease (OR 1.53 [1.13–2.09]; *p*=0.006). A decreased likelihood of early AC initiation was seen in patients with increased age (OR 0.70 [0.51–0.97]; *p* = 0.036), a Comorbidity Score of 1 (OR 0.70 [0.51–0.97]; *p* = 0.032), and treatment at an Academic or Research Institution (OR 0.43 [0.20–0.92]; *p* = 0.030) [15].

In 2020, the University of Miami Miller School of Medicine published a retrospective analysis that looked at 284 patients from 2004 to 2014 in the NCDB that had newly diagnosed pT2-T4, N0, M0 MIBC that did not receive any NAC prior to RC. They separated patients into cohorts that received no AC, AC within 45 days, and AC after 45 days. The median survival time in patients who received AC within 45 days after RC was the highest, at 4.3 years (2.9–5.7). Patients that received AC within 45 days also had the greatest median 5-year OS (47.0%; [40.6–53.2]), 3-year OS 55.1% (48.9–60.8), and 1-year survival of 85.9% (81.3–89.4). In patients who received AC after 45 days, they found a decrease in overall median survival by 1.5 years (2.8 y [2.1–3.2]), and median survival by 9.5% at 5 years (37.5% [31.4–43.7]), 8.6% at 3 years (46.5% [40.3–52.4]), and 0.3% (85.6% [2.1–3.2)]) at 1 year. Receipt of AC after 45 days and no AC predicted worse OS compared to AC within 45 days (1.27 [1.02–1.59], *p* = 0.033 and 1.41 [1.12–1.78], *p* = 0.003). They also showed that patients who received no AC compared to AC after 45 days did not differ significantly (1.11; 0.89–138). A significant number of patients (53.2%) who received AC within 45 days had a hospital stay of ≤ 7 days after RC, compared to those who received AC after 45 days (44.0%) or no AC at all (43.0%; *p* = 0.0369) [16].

## 4. Discussion

The rationale behind perioperative chemotherapy utilization in patients with MIBC is as follows: the presence of occult micrometastases cannot be accounted for in those with locally advanced disease, and surgery alone is insufficient for cancer control [17]. Current guidelines for MIBC call for the use of NAC before surgical consolidation. In certain situations, acquiring precise pathologic data before administering can make AC an attractive option. Pathologic staging systemic treatment and reducing the time to surgery can predict prognosis and the absolute OS benefit of AC appears to be most evident in patients with at least cT3 disease [2,18]. The primary limitation of adjuvant therapy is that delays in administration can occur secondary to poor postoperative healing or complications, leading to the propagation of potential micrometastases, earlier recurrence, and reduction in cancer-specific survival. The recent approval of IO for patients with adverse surgical pathology makes adjuvant treatment a possibility, even in those who have received NAC prior to RC. CheckMate 274, a multicenter randomized trial, included patients with MIBC who underwent RC and were administered either adjuvant nivolumab (*n* = 353) or placebo (*n* = 356). The median DFS in the nivolumab group was 20.8 months (16.5–27.6) compared to 10.8 months (8.3–13.9) in the control group. Patients alive and disease free at six months were 74.9% in the nivolumab group and 60.3% in the placebo cohort (HR for disease recurrence or death: 0.70; 98.22% CI [0.55–0.90]; *p* < 0.001) [5]. IMvigor010, a multicenter randomized trial, either administered adjuvant atezolizumab (*n* = 406) or assigned patients to observation (*n* = 403) after they received a RC for MIBC (90%) or a nephroureterectomy with lymph node dissection (10%) for upper tract urothelial carcinoma. The median DFS in the atezolizumab group was 19.4 months (15.9–24.8) compared to 16.6 (11.2–24.8) for observation (HR 0.89 [0.74–1.08]; *p* = 0.24) [19]. Although these studies show great promise, OS data is not yet mature in adjuvant IO literature.

The mean hospital stay after RC at most centers is as high as 9 to 11 days and even as high as 17.4 in a recent European series [8]. In a recent trial, the robot-assisted radical cystectomy with intracorporeal urinary diversion versus open radical cystectomy (ROC) suggests that patients who undergo a robotic-assisted RC compared to open spend less time in the hospital within the first 90 days following surgery (95% CI, 0.5–3.85; *p* = 0.01) [20]. In contrast, other studies have suggested that there are no significant differences in complications and length of stay (LOS) in the hospital in a robotic versus open radical cystectomy; the latter supports the notion that perioperative management is the most critical piece to patients recovering faster and decreasing LOS in the hospital [8]. The University of Southern California initiated an ERAS protocol in 110 patients after RC and was able to decrease their median LOS from 8 to 4 days [8]. A systematic review and meta-analysis that looked at ERAS compared to the standard of care following cystectomy and showed a lower overall complication rate (RR 0.85 [0.74–0.97]; *p* = 0.017; I^2^ = 35.6%) and a shorter LOS (standardized mean difference: −0.87 [−1.31 to −0.42]; *p* = 0.001; I^2^ = 92.8%) [9]. Implementing ERAS into clinical practice can lead to faster recovery after RC and allow patients who need AC to receive it quicker, ultimately improving survival.

Despite efforts to improve recovery and shorten the length of stay with the adoption of ERAS protocols, RC remains associated with significant morbidity [9]. Donat et al., published a single-institution retrospective analysis of a prospective complication database to delineate the frequency of adverse (AE) within a 90-day postoperative period [21]. In this study, 57% of the noted complications (347 of 611), grades 2–5 AEs, occurred within this 90-day cutoff and 38% within six weeks of surgery. Overall, 30% of patients within this cohort could have experienced delays in receiving AC within this 90-day interval secondary to postoperative complications. The most significant predictors of AC delay were advanced age, higher EBL, higher ASA, prior abdominal surgery, prior radiation, and type of urinary diversion [21]. Grunewald et al., failed to substantiate whether the 90-day cutoff often employed in AC studies after post-RC is optimal.

Sternberg et al., published an intention-to-treat analysis of a multicenter phase 3 RCT in 2015, which aimed to compare survival trends in patients with pT3-T4 or N+ M0 MIBC who received immediate AC versus those who received deferred chemotherapy at relapse [22]. Patients were randomized to the two groups mentioned above within 90 days of surgery (the median time to randomization was 63 days). The median OS was 6.74 years and 4.60 years in the immediate and deferred groups, respectively (HR 0.78 [0.56–1.08]). While PFS was seemingly improved in the immediate AC group (HR 0.54 [0.40–0.73]), the duration of survival after treatment, which is arguably more clinically relevant, was significantly shorter in this group (HR 1.45 [1.02–2.07]). Time from cystectomy to randomization did not affect OS (≤30 days, 31–60 days, and >60 days had overlapping HRs). Interestingly, a subgroup analysis of 86 patients with negative lymph nodes at baseline revealed improved OS in those receiving immediate AC (HR 0.37 [0.16–0.83]), suggesting that certain MIC patient subpopulations can benefit from immediate consolidative surgery and systemic treatment [22].

Three studies in this review showed a statistically significant survival benefit in patients post-RC when they received early AC compared to delayed chemotherapy. Four studies showed an increased survival benefit when receiving AC within 90 days (Table 4). Grunewald et al., failed to substantiate whether the often-employed 90-day cutoff seen in AC studies after RC is optimal. In contrast, Corbett and Jue et al., suggest that systemic treatment, even after the often-proposed cutoff of 90 days, still provides a significant survival benefit [14,15,16]. Collectively, they support the idea that there are survival benefits when AC is given as soon as safely possible, as they showed a benefit at earlier intervals within 30, 45, and 60 days.

As noted in our review, chemotherapy regimens were heterogeneous in terms of types of drugs and number of cycles used, limiting the generalizability of the data. Prospective literature for AC in MIBC is sparse, and many studies have poor accrual or are underpowered [23]. The need for such high-impact studies on this topic persists and should also be explored for adjuvant IO to see if similar survival benefits can be shown when administering IO earlier. At this time, a safe and ethical approach would be to utilize ERAS protocols and to administer AC as early in the postoperative period as possible, given the known 9–11% absolute survival benefit at five years attributable to AC [6,7].

## 5. Conclusions

Data regarding the importance of postoperative timing for AC in MIBC is heterogeneous. Patients appear to show a survival benefit the earlier they receive AC, albeit a benefit persists even after receiving delayed AC or systemic treatment at relapse. Therefore, the general adoption of the 90-day window is arbitrary. Additional adequately powered prospective studies are needed to make a definitive conclusion regarding this topic. Nonetheless, given the known 9–11% absolute survival benefit at five years attributable to early TTAC in this cohort, an ethical approach would be maximizing recovery after surgery using ERAS protocols and administering AC as early as possible in the postoperative period.

## Figures and Tables

**Figure 1 cancers-14-05644-f001:**
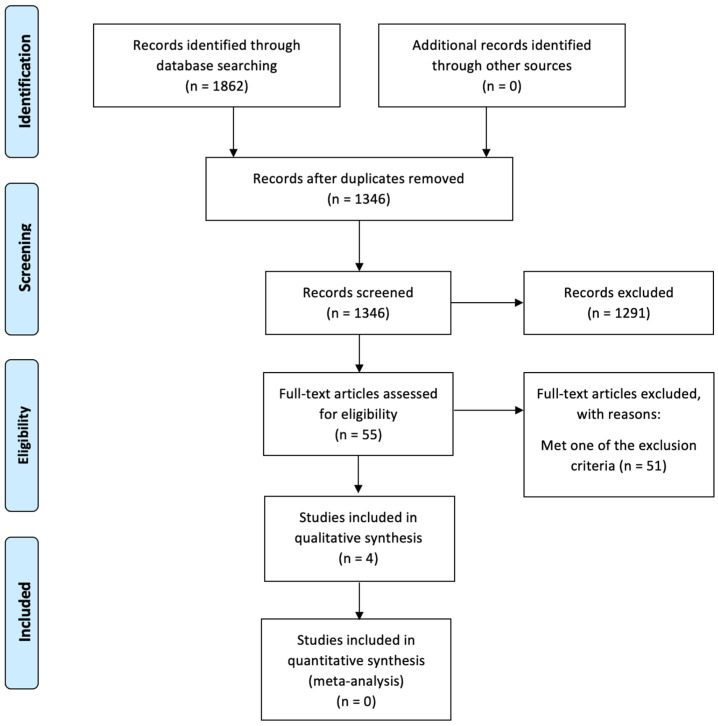
PRISMA flow diagram. Legend: The PRISMA diagram details the search and selection process applied during the systematic review.

**Table 1 cancers-14-05644-t001:** Data Collection Type.

Study Author	Year Published	Sample Size	Data Collection	Level of Evidence	Bias Risk
Corbett et al.	2019	3590	Retrospective	2b	Moderate
Jue et al.	2020	284	Retrospective	2b	Moderate
Booth et al.	2014	2944	Retrospective	2b	Some concerns
Grunewald et al.	2020	238	Retrospective	2b	Moderate

**Table 2 cancers-14-05644-t002:** Summary of Studies.

Study Author	Days between RC and AC	Site of Study	Database	Patients (*n*)	Years Analyzed
Corbett et al.	No AC, 31–60 d, 61–90 d, 91–180 d	US	NCDB	3590	2006–2013
Jue et al.	No AC, AC < 45 d, AC > 45 d	US	NCDB	284	2004–2014
Booth et al.	Mean/median TTAC—10 and 9 wks, respectively	Ontario	Ontario Cancer Registry	2944	1994–2008
	23% (120/533) started ACT > 12 wks s/p RC				
Grunewald et al.	Start of AC (S-AC) < 90 d vs. S-AC > 90 d	28 International centers	Retrospective International Study of Invasive/Advanced Cancer of the Urothelium (RISC) database	238	2005–2012

**Table 3 cancers-14-05644-t003:** Demographics.

Study	Age (Mean)	Race	Gender	pT Stage	pN Stage	Other
Corbett et al.						
No AC (*n* = 2581)	70.5	91.4% (*n* = 2360) white, 5.5% (*n* = 142) black, 3.1% (*n* = 79) other	91.4% (*n* = 2360) white, 5.5% (*n* = 142) black, 3.1% (*n* = 79) other	<2 = 5.9% (*n* = 151), 3 = 75% (*n* = 1937), 4 = 19.1% (*n* = 493)	0 = 72% (*n* = 1860), 1 = 13% (*n* = 335), >2 = 15% (*n* = 386)	Margins: Negative 89% (*n* = 2291), positive = 11% (*n* = 290)
31–60 d (*n* = 538)	64.2	93% (*n* = 500) white, 4.8% (*n* = 26) black, 2.2% (*n* = 12) other	77% (*n* = 414) male	pT stage: <2 = 10% (*n* = 54), 3 = 63% (*n* = 341), 4 = 27% (*n* = 143)	pN stage: 0 = 34% (*n* = 183), 1 = 26% (*n* = 141), >2 = 40% (*n* = 214)	Margins: Negative 80% (*n* = 429), positive = 20% (*n* = 109)
61–90 d (*n* = 321)	65.8	88.2% (*n* = 283) white, 8.4% (*n* = 27) black, 3.4% (*n* = 11) other	75% (*n* = 241) male	pT stage: <2 = 13% (*n* = 43), 3 = 65% (*n* = 208), 4 = 22% (*n* = 70)	pN stage: 0 = 42% (*n* = 135), 1 = 21% (*n* = 68), >2 = 37% (*n* = 118)	Margins: Negative 84% (*n* = 270), positive = 16% (*n* = 51)
90–180 d (*n* = 150)	65.1	90% (*n* = 135) white, 4.7% (*n* = 7) black, 5.3% (*n* = 8) other	74% (*n* = 111) male	pT stage: <2 = 8.7% (*n* = 13), 3 = 66% (*n* = 99), 4 = 25.3% (*n* = 38)	pN stage: 0 = 47% (*n* = 70), 1 = 26% (*n* = 39), >2 = 27% (*n* = 41)	Margins: Negative 87% (*n* = 130), positive = 13% (*n* = 20)
**Jue et al.**						
All patients	62.4 (9.7)	93% (*n* = 792 white), 3.1% (26) black, 2.1% (18) other, 1.9% (16) unknown	87.8% (748) male	2 = 18.2%, 3 = 59.2%, 4 = 22.7%	-	
≤45 d	62.4 (9.4)	92.3% (*n* = 262 white), 3.5% (26) black, 2.1% (6) other, 2.1% (6) unknown	87% (247) male	2 = 19%, 3 = 57%, 4 = 23.9%	-	
>45 d	62.5 (9.4)	93.3% (*n* = 265 white), 2.8% (8) black, 2.1% (6) other, 1.8% (5) unknown	89.1% (253) male	2 = 16.5%, 3 = 61.3%, 4 = 22.2%	-	
No Chemo	62.4 (10.3)	93% (*n* = 265 white), 2.8% (8) black, 2.1% (6) other, 1.8% (5) unknown	87.3% (248)	2 = 19%, 3 = 59.2%, 21.8%	-	
Booth et al.	60% >70	-	77% (513) male	71% locally advanced T3/T4	26% pN+	NAC + AC—82% (253/308) cisplatin, 14% (43/308) carboplatin, 4% (12/308) other, 1% (670) NAC + AC; 308/670 NAC + AC = 82% (253/308) cisplatin, 14% (43/308) carboplatin, 4% (12/308) other
Grunewald et al.	63.4 (9.5)	-	83.5% male (197)	76.1% (181) > pT3/T4	66.4% (158) = pN+	71% (*n* = 169) cisplatin-based AC; majority of patients receiving 4 cycles (median 4, interquartile range = 1) 87% (*n* = 207) S-AC < 90 d s/p RC; median S-AC = 57 d (range 10–331; interquartile range = 32.8)

**Table 4 cancers-14-05644-t004:** Overall Survival.

Study	Time	Median 5-Yr Survival	Median OS	HR	*p* Value	Adjusted HR	*p* Value
**Corbett et al.**	No AC (reference)		26.9 months				
	Overall AC treated patients		35.8 months				
	AC 31–60 d		33.5 months	0.6 (0.52–0.69)	<0.001		
	AC 61–90 d		37.8 months	0.62 (0.53–0.74)	<0.001		
	AC 91–180 d		36.1 months	0.69 (0.55–0.87)	0.002		
**Jue et al.**	No AC	41.2% (35.0–47.2)	2.2 yr (1.8–3.4)				
	AC < 45 d	47% (40.6–53.2)	4.3 yr (2.9–5.7)				
	AC > 45 d	37.5% (31.5–43.7)	2.8 yr (2.1–3.2)				
	Overall	41.2% (35.0–47.3)	2.9 (2.5–3.4)				
	>45 d vs. ≤45 d			1.25 (1.01–1.56)	0.044	1.27 (1.02–1.59)	0.033
	No AC vs. ≤45 d			1.36 (1.09–1.70)	0.006	1.41 (1.12–1.78)	0.003
	No AC vs. >45 d			1.09 (0.88–1.35)	0.442	1.11 (0.89–1.01)	0.348
**Booth et al.**	AC overall	29% (25–33)					
	NAC overall	25% (17–34)					
	AC < 12 wks (median 9 wks)	33%		Ref			
	AC > 12 wks	22%	Inferior	1.28 (1.00–1.62)			*p* value trend—0.046
**Grunewald et al.**	AC < 90 d (median 57 d)	(45–87)	73 months	1.285 (0.682–2.418)	0.438		
	AC > 90 d	(15–n.a.)	48 months	1.285 (0.682–2.418)	0.438		

## Data Availability

Not applicable.

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
