# Peer review of "Does Time to Adjuvant Chemotherapy after Radical Cystectomy Affect Survival in Muscle Invasive Bladder Cancer? A Systematic Review"

_cancers, 2022, doi:10.3390/cancers14225644_

Round 1
Reviewer 1 Report
In the present article Kronstadt et al. present a systematic review on the impact of the delay of adjuvant chemotherapy after radical cystectomy.
This is an interesting topic and the article is well written. However, I have some concerns regarding this article:
1. Is a postoperative chemotherapy administered >12weeks after surgery still an adjuvant approach?
2.The administered regimen are heterogenous (Carbo vs Cis). This needs to discussed.
3. Why was the Sternberg trial not included in the analyses? It is discussed but should it not be included in the analyses as it is prospective?
Finally, the present is not suited for publication in its present form.
Reviewer 2 Report
I appreciate the authors for their systemic review on this important topic. Very well written and analysed.
The study is limited by 4 studies and mostly retrospective data, which obviously is a limitation. However, there are no prospective studies and mostly RCT evaluating novel therapies aim to start Adj treatment with in 90-120 days of surgery.
Author Response
Thank you for your time in reviewing our paper and for the excellent feedback. We agree that this is a limitation of our paper, which was included in the last paragraph of our discussion.
Reviewer 3 Report
The paper of Kronstedt et al. consists of 9 pages text, 1 figure, 4 tables and 24 references. The paper is a systematic review of the literature regarding the role of adjuvant chemotherapy after radical cystectomy in musckle invasive bladder cancer. Main goal oft he evaluation is the important clinical conclusion, that chemotherapy hast o follow as early as possible. The focus of time to adjuvant treatment is new and differs from papers oft he ABC group. Based on this, I accept as a reviewer this paper in the present form. In the future similar papers have to discuss and consider the role of adjuvant immuno oncology regarding this topic.
Author Response
Thank you for your time in reviewing our paper and for the comments. We also agree with your statement and made sure it was something that was discussed at the end of our paper in the last paragraph of the discussion section.